# CCR8 as a Therapeutic Novel Target: Omics-Integrated Comprehensive Analysis for Systematically Prioritizing Indications

**DOI:** 10.3390/biomedicines11112910

**Published:** 2023-10-27

**Authors:** Nari Kim, Mi-Hyun Kim, Junhee Pyo, Soo-Min Lee, Ji-Sung Jang, Do-Wan Lee, Kyung Won Kim

**Affiliations:** 1Department of Medical Science, Asan Medical Institute of Convergence Science and Technology, Asan Medical Center, University of Ulsan College of Medicine, Seoul 05505, Republic of Korea; nari.kim.0908@gmail.com; 2Research Institute, Trial Informatics Inc., Seoul 05544, Republic of Korea; mhkimasan@gmail.com; 3College of Pharmacy, Chungbuk National University, Cheongju 28644, Republic of Korea; stdpjh@naver.com; 4Samjin Pharmaceutical Co., Ltd., Seoul 04054, Republic of Korea; soominlee@samjinpharm.co.kr; 5Biomedical Research Center, Asan Institute for Life Sciences, Asan Medical Center, Seoul 05505, Republic of Korea; etmira8787@gmail.com; 6Department of Radiology and Research Institute of Radiology, Asan Medical Center, University of Ulsan College of Medicine, Seoul 05505, Republic of Korea; ldw841115@gmail.com

**Keywords:** anti-cancer, target identification, comprehensive analysis, multi-omics analysis, drug target, C-C chemokine receptor type 8 (CCR8), priority indication

## Abstract

Target identification is a crucial process in drug development, aiming to identify key proteins, genes, and signal pathways involved in disease progression and their relevance in potential therapeutic interventions. While C-C chemokine receptor 8 (CCR8) has been investigated as a candidate anti-cancer target, comprehensive multi-omics analyzes across various indications are limited. In this study, we conducted an extensive bioinformatics analysis integrating genomics, proteomics, and transcriptomics data to establish CCR8 as a promising anti-cancer drug target. Our approach encompassed data collection from diverse knowledge resources, gene function analysis, differential gene expression profiling, immune cell infiltration assessment, and strategic prioritization of target indications. Our findings revealed strong correlations between CCR8 and specific cancers, notably Breast Invasive Carcinoma (BRCA), Colon Adenocarcinoma (COAD), Head and Neck Squamous Cell Carcinoma (HNSC), Rectum adenocarcinoma (READ), Stomach adenocarcinoma (STAD), and Thyroid carcinoma (THCA). This research advances our understanding of CCR8 as a potential target for anti-cancer drug development, bridging the gap between molecular insights and creating opportunities for personalized treatment of solid tumors.

## 1. Introduction

C-C chemokine receptor type 8 (CCR8) is a cell surface receptor that belongs to the G protein-coupled receptor (GPCR) family [1]. It is a protein expressed on the surface of various immune cells, including regulatory T cells (Tregs) [1]. Tregs have the ability to suppress the activity of other immune cells, including cytotoxic T cells and natural killer cells, which are responsible for recognizing and attacking tumor cells [2]. In peripheral tissues, Resting conventional T cells (T conv cells) can differentiate into inducible regulatory T cells (iTreg) in the presence of specific cytokines such as TGF-β and IL-2 [3]. The CCR8 receptor on the surface of Tregs is then upregulated by local cytokine and chemokine signaling within the tumor site. This immunosuppressive effect of Tregs at the tumor site can indeed weaken the anti-tumor immune response, making them an important target for therapeutic intervention in cancer immunotherapy [4]. However, such Treg-targeting cancer immunotherapies occasionally induce immunopathology and autoimmunity as adverse effects [5]. Several studies have suggested that targeting CCR8 has the potential to be more specific in anti-tumor activity than other current approaches to Tregs depletion [2,6]. CCR8 is known to play a crucial role in recruiting Tregs to the tumor site, fostering an immunosuppressive environment that aids tumor escape [4]. By inhibiting CCR8, it is possible to disrupt this recruitment process, potentially enhancing anti-tumor immune responses and suppressing tumor growth [2]. Recent studies have suggested that disruption of CCR8 function using anti-CCR8 antibodies reduces the accumulation of Treg cells in tumors and disrupts their immunosuppressive function [7]. Indeed, certain studies have shown that targeting CCR8+ T cells through depletion therapy using anti-CCR8 monoclonal antibodies (mAbs) in mice can trigger tumor-specific immune responses against tumors, without causing autoimmune reactions or immune responses within the tumor microenvironment [8].

In the field of drug development, target identification is to identify proteins, genes, and signal pathways that play an important role in disease progression, to determine what role the target plays in the disease development mechanism and in which patient population pharmacological modulation could be effective [9,10]. Comprehensive analysis of targets is essential to establish their relevance, validate their role, assess their druggability, predict outcomes, and develop the basis for potential therapeutic interventions [11,12]. Additionally, in the early stages of drug development, bioinformatic target identification provides meaningful insights into disease mechanisms based on extensive datasets [13]. Although several studies have explored the candidate CCR8 as an anti-cancer target, analyzes based on various multi-omics databases are still not common, so a comprehensive investigation of therapeutic potential across the spectrum of indications is needed. Indeed, numerous studies have focused on exploring CCR8 as a potential approach for anti-cancer drug development, but studies of its association with T-cell lymphoma, a type of hematological cancer, have received more attention than solid cancer [14,15]. Recently, the prospect of Treg-mediated cancer immunotherapy targeting CCR8 has gained significant attention, leading biopharmaceutical companies to make various efforts in developing anti-CCR8 agents for cancer treatment. Most of the anti-ccr8 pipelines being developed in the preclinical or clinical trial stages are monoclonal antibodies, which have a mechanism to kill tumors after selective targeting through antibody-dependent cytotoxicity (ADCC) action. Antibody drugs, including IPG7236 [14] (Immunophage Biomedical Co., Ltd., Nanjing, China), S-531011 [16] (Shionogi Pharma Co., Ltd., Osaka, Japan), BMS-986340 (Bristol Myers Squibb Co., New York, NY, USA), LM-108 (LaNova Medicines Ltd., New York, NY, USA), SRF-114 (Vaccinex, Inc., New York, NY, USA), have entered phase 1/2 clinical trials and are recruiting patients. However, many of the anti-cancer drugs targeting CCR8 in the current clinical trial stage are being studied for solid tumors without specific predefined indications, so there is a need to determine indications based on the mechanism of the disease.

Our study aims to establish CCR8 as a promising and potential target for anti-cancer drug development by performing a comprehensive bioinformatics analysis covering genomics, proteomics and transcriptomics. By exploring potential indications and molecular interactions, we provide valuable insights to assist in the creation of more effective, personalized treatments for solid tumors. This research bridges the gap between molecular understanding and clinical application, advancing the field of anti-cancer drug development.

## 2. Materials and Methods

### 2.1. Framework for Omics-Integrated Analysis

We used cancer-related data resources for multi-omics analysis to prioritize target indications based on molecular pathways and gene function, tissue-specific distribution, correlation of CCR8 gene with immune cells, and patient survival outcomes.

In this study, we present a comprehensive analytical framework to demonstrate CCR8 as a promising anti-cancer drug target through a multi-omics approach (Figure 1). Our comprehensive analysis includes data collection, gene function analysis, differential gene expression profiling, immune cell infiltration analysis, and strategic prioritization of target indications. All TCGA cancer abbreviations are summarized in Table 1.

### 2.2. Gene Function Analysis

Understanding the function of genes is important for identifying potential effects of genes in disease mechanisms by providing insight into the role of genes in different biological contexts. In terms of gene ontology, the molecular function (MF) class describes the activities of the gene product, and the cellular component (CC) refers to where the gene product is active. The biological process (BP) refers to the pathways and processes to which the gene product’s activity contributes [17]. Pathway maps sourced from NDEx Query were initially compiled based on the published literature. Extracted pathway maps underwent a review process to eliminate redundant processes, resulting in a concise summary of their roles within biochemical signaling pathways. We retrieved gene function information and molecular pathways, and protein–protein interactions for the search term “CCR8” from the Web Gene Ontology Resource Database (http://geneontology.org) (accessed on 30 August 2023) and the NDEx Query database version 1.4 (https://www.ndexbio.org/iquery/) (accessed on 30 August 2023) [18]. Protein–protein interactions (PPIs) with interaction maps were retrieved from the STRING database web server system version 12.0 (https://string-db.org) (accessed on 30 August 2023) [19], which incorporates both known and predicted PPIs [20].

Then, we investigated the complex network of interactions between CCR8 and a diverse set of chemokines by protein–protein interaction network analysis [21]. The degree of protein–protein interaction was calculated using a combined score based on various sources of evidence to estimate the reliability and significance of predicted protein–protein interactions [22].

### 2.3. Target Gene Expression Profiling

To analyze the potential of CCR8 as a new target for anti-cancer drug development, we investigated gene expression levels across various cancer and normal tissues as well as various cell types. First, we searched the CCR8 gene in 33 TCGA tumor types from the TIMER web source version 2.0 (http://timer.cistrome.org/) (accessed on 30 August 2023) [23] to obtain information about differential expression between tumors and adjacent normal tissues. The distribution of gene expression levels by cancer type is shown using boxplots. Statistical significance calculated by Wilcoxon test is annotated with stars (*: *p*-value < 0.05; **: *p*-value < 0.01; ***: *p*-value < 0.001) [24]. Secondly, utilizing the GEPIA platform version 2.0 (http://gepia2.cancer-pku.cn/) (accessed on 30 August 2023) [25], we focused on cancer-specific gene expression patterns across different cancer types. We then presented overall CCR8 protein expression levels for each of the 44 organs across normal tissues based on knowledge-based annotations obtained from the Human Protein Atlas version 23.0 (https://www.proteinatlas.org) (accessed on 30 August 2023) [26]. Thirdly, the TIMER 2.0 database facilitates the evaluation of CCR8 expression within tumor-infiltrating immune cells, revealing its potential role in the tumor microenvironment [27]. The integration of these datasets and online resources has led to a comprehensive understanding of the differential expression of CCR8 in cancer and normal tissues, its potential as a cancer-specific biomarker, and its involvement in immune cell populations within the tumor environment.

### 2.4. Immune Cell Infiltration Analysis

Immune cell infiltration analysis is a key method for interpreting the complex relationship between immune cells and their microenvironment, offering insights into disease progression, treatment responses, and potential immunotherapeutic approaches. This analysis will encompass the correlations between CCR8 subunit expression levels and tumor immune infiltration levels (B cells, CD4 T cells, CD8 T cells, neutrophils, macrophages, and dendritic cells) and investigate the impact of CCR8 expression on regulatory T cell (Treg) expression. To investigate the association between cancer-associated fibroblasts (CAFs) and the expression levels of specific genes, we performed TIDE, xCell, MCPcounter, and EPIC analyzes provided by TIMER web solutions.

The correlation between the level of CCR8 expression and the level of infiltration of each tumor-infiltrating immune cell subtypes (Activated dendritic cell, M2 macrophage, Myeloid-derived suppressor cells (MDSC), Tregs) was analyzed using the Spearman correlation and its coefficient, rho [28]. The rho value represents the strength and direction of the linear relationship between the CCR8 gene expression level and tumor-infiltrating immune cells. We generated a heatmap table of Spearman correlations between the expression of input genes and the abundance of immune cell types. The strength of the correlation coefficient, rho, was graded as strongly positive (0.70 to 1.00), moderately positive (0.30 to 0.70), weak (0.10 to 0.30), negligible (−0.10 to 0.10), moderately negative (−0.70 to −0.30), or strongly negative (−1.00 to −0.70) [28]. We conducted this analysis using the Tumor Immunity Estimation Resource (TIMER 2.0) (https://cistrome.shinyapps.io/timer/) (accessed on 30 August 2023) [23].

### 2.5. Prognostic Value Analysis

GEPIA 2.0 (http://gepia2.cancer-pku.cn/) (accessed on 30 August 2023) is an online analysis solution that analyzes gene expression based on 8587 normal and 9736 tumor samples from the TCGA and GTEx datasets using the output of a standard processing pipeline for RNA sequencing data [26]. From the GEPIA database, we extracted patients’ data from TCGA datasets which include the RNA sequencing expression levels of CCR8 and overall survival data in 33 distinct cancer types [26,27]. Survival analysis comparing groups with high and low levels of gene expression is also widely used to assess the clinical significance of specific genes [24]. The cut-off of high level and low level of gene expression was determined to be at 50% of patients. We included the Treg marker gene *FOXP3* as a control to demonstrate that CCR8-positive Treg expression is indeed a factor causing immune suppression.

Based on the extracted datasets, the prognostic value of CCR8 expression level on overall survival was analyzed using univariate and multivariate Cox proportional hazard models. Kaplan–Meier survival curves were also generated. The overall survival of patients with high CCR8 expression and low CCR8 expression was compared using the Log-rank test.

A multivariate Cox proportional hazard model was constructed based on CCR8 levels with various covariates including cancer stages, B cell, CD4+ T cell, CD8+ T cell, macrophage, neutrophil, and dendritic cell counts. This analysis provides valuable insight into the complex interactions between immune cell subsets, CCR8 levels, and cancer stage in determining patient survival across multiple cancer types. If there were no cancer stage data in a cancer type, we removed the cancer stage from the covariates. The R codes for the multivariate Cox proportional hazard model for each tumor type within the TIMER web solution are as follows:Model: Survival outcome (by cancer type) ~ Stage + B cell + CD8 T cell + CD4 T cell + Macrophage + Neutrophil + Dendritic cell + CCR8 level

### 2.6. Prioritization of Target Indications

The comprehensive analysis utilizing multiple types of data is indeed aimed at ultimately determining the appropriate target indications for a particular intervention or treatment strategy. We performed step-by-step analysis of the potential of the CCR8 gene as an anti-cancer drug target using an omics-integrated comprehensive analysis framework. Therefore, we developed a summary table to integrate the individual results and evaluate the correlation of CCR8 with various cancer types. In the comprehensive evaluation, a strong correlation was defined as when gene expression analysis, immune infiltrating cell analysis, and prognosis evaluation were all applicable (Table 2).

## 3. Results

### 3.1. Gene Function Analysis

The biological processes of CCR8 are involved in the immune response, cell adhesion, the G protein-coupled receptor signaling pathway, the chemokine-mediated signaling pathway, positive regulation of cytosolic calcium ion concentration, and chemotaxis. The molecular functions of CCR8 include coreceptor activity, C-C chemokine receptor activity, and chemokine receptor activity. These functions reflect its role as a cell surface receptor that binds to specific chemokines and participates in cell signaling processes, including immune responses and cell migration [29]. Detailed information about gene functions is presented in Table 3.

CCR8 and its ligand CCL1 play an important role in regulating the recruitment and function of Tregs within the tumor microenvironment (TME) [30]. CCR8 is a receptor expressed on the surface of Tregs, while CCL1 is a chemokine secreted by various cells within the TME. Binding of CCL1 to CCR8 on Tregs promotes the migration of these regulatory immune cells to the site of CCL1, which is often the tumor site (Figure 2). The influx of Tregs into the TME may lead to expansion and activation of the TME, contributing to immunosuppression and immune tolerance within the tumor and ultimately interfering with effective anti-tumor immune responses. Signaling by CCR8–CCL1 interaction promotes the migration of Treg cells to the site of inflammation and enhances the suppressive function of Treg cells, contributing to suppressing the immune response at the tumor site. Accordingly, understanding the CCR8–CCL1 axis is essential to develop strategies to modulate Treg activity in cancer treatment.

The protein–protein interaction analysis revealed a significant and complex network of interactions between CCR8 and a diverse set of chemokines, including C-C Motif Chemokine Ligand 1 (CCL1), CCL17, CCL18, CCL22, CCL4, CCL8, CCL16, CCL20, CCL5, and CCL2. (Figure 3) [15,29]. Their combined scores were greater than 0.9. Of these, the CCL1 showed the highest score of 0.999, followed by CCL17 (0.998) and CCL18 (0.997).

### 3.2. Target Gene Expression Profiling

As shown in Figure 4a, CCR8 differentially expressed gene (DEG) analysis revealed increased expression in cancer tissues compared with normal tissues. In particular, statistically significant gene expression increases were observed in several cancer types, including BLCA, BRCA, COAD, ESCA, HNSC, KIRC, LIHC, LUAD, LUSC, SKCM, STAD and UCEC, highlighting their potential relevance for various cancers. Figure 4b also highlights the prominent expression of CCR8 in the thymus among body tissues, suggesting a potential role in thymic function for production and maturation of T cells. Notably, CCR8 exhibited high expression in key areas such as the gastrointestinal tract, lungs, spleen, lymph nodes, and tonsils. These regions are characterized by active immune reactions resulting from interactions between self and foreign elements. Also, it was found that CCR8 was highly expressed in T-reg among blood and immune cells (Figure 4c).

### 3.3. Immune Cell Infiltration Analysis

#### 3.3.1. Correlation of CCR8 Level with Tumor Immune Cell Infiltration Level

The identified cancers with significant correlations between CCR8 and immune cell infiltration are BLCA, BRCA, COAD, HNSC, and KIRC (Figure 5). The x-axis represents the immune cell infiltration level, while the y-axis represents the CCR8 expression level on Treg.

The negative correlation between CCR8 and tumor purity implies that as CCR8 expression on Treg increases in these carcinomas, the tumor purity decreases. Tumor purity refers to the proportion of cancerous cells in the tumor microenvironment [30]. A negative correlation with tumor purity suggests that high CCR8 expression on Treg is associated with a higher presence of non-cancerous cells, such as immune cells, in the tumor. Also, CCR8 shows positive correlations with a variety of immune cell types, including B cells, CD4 T cells, CD8 T cells, neutrophils, macrophages, and dendritic cells. These positive correlations indicate that higher CCR8 expression on Treg is associated with increased infiltration of these immune cells into the tumor microenvironment. Immune cell infiltration is often regarded as beneficial in cancer treatment; however, immune cells may not function effectively when there is an abundance of regulatory T cells (Tregs) present [31].

#### 3.3.2. Correlation Analysis of CCR8 Expression Levels with Levels of Tumor-Infiltrating Immune Cell Subgroups

The strong correlations observed between CCR8 expression and specific immune cell subgroups suggest the druggable potential of anti-CCR8 agents. The correlation with Tregs indicates that targeting CCR8-expressing Tregs could potentially suppress immunosuppressive effects within the tumor microenvironment. Additionally, the high correlation with dendritic cells implies the potential to enhance anti-tumor effects through ADCC action. (Table 4, Figure 6).

Considering the tumor-infiltrating Tregs, a statistically significant correlation between the level of CCR8 expression and the level of infiltration of Tregs was observed in certain but not all cancer types. In particular, very strong and strong correlations were observed in BLCA, BRCA, CESC, CHOL, COAD, ESCA, HNSC, LIHC, LUAD, LUSC, MESO, PAAD, PRAD, READ, SKCM, STAD, THCA. This highlights that the infiltrating Tregs express CCR8 significantly only in specific cancers where Treg-mediated immune suppression is pronounced.

Between the three Treg gene markers, CTLA4 (r = 0.71, *p* < 0.001) and *FOXP3* level showed a very strong correlation with CCR8 expression level (r = 0.82, *p* < 0.001), while IL-10 (r = 0.69, *p* < 0.01), STAT5B (r = 0.67, *p* < 0.01) and TGFB1 (r = 0.53, *p* < 0.001) showed a strong correlation with CCR8 (Figure 7). In contrast, *TP53*(r = 0.23, *p* < 0.001), the control gene, showed very low or negative correlation with CCR8 expression. These findings indicate that increased CCR8 expression in the tumor microenvironment reflects an increase in *FOXP3* + Tregs and contributes to immunosuppression.

#### 3.3.3. Cancer-Associated Fibroblast (CAF) Level Correlations with CCR8 Expression Level

Among cancers, moderate correlations between CCR8 expression and CAF levels and were observed in COAD (0.572), READ (0.514), THCA (0.503), HNSC-HPV− (0.451), PAAD (0.446), etc., (Figure 8). Such positive correlations suggest that targeting CCR8 could potentially interfere with CAF-mediated immunosuppression and enhance anti-tumor immune responses.

### 3.4. Prognostic Value Analysis

#### 3.4.1. Univariate Analysis

As a result of a search on the TIMER 2.0 web source, 4 out of 33 cancers showed significant differences in overall survival between the high and low expression groups (Table 5). The order of prognostic effect of CCR8 for overall survival based on the hazard ratio was as follows: GBM (hazard ratio, 1.19), KIRP (2.20), LGG (1.80), and UVM (4.50). In the case of FOXP3, similar trends were observed across the same cancer types: GBM (1.80), KIRP (1.90), LGG (1.50), and UVM (2.60) (Figure 9).

#### 3.4.2. Multivariate Analysis

The calculated hazard ratios (HR) along with their corresponding 95% confidence intervals (CI) and *p*-values from the Cox proportional hazards analysis indicate the impact of CCR8 expression on the overall survival of various cancer types: BRCA, COAD, HNSC, KICH, LIHC, MESO, PAAD, and OV (Table 6). These findings imply that CCR8 expression may potentially have an impact in regulating tumor immunity and prognosis in various cancer types. Components of tumor immunity (B cells, CD4+ T cells, CD8+ T cells, macrophages, neutrophils, and dendritic cells) have been identified as major immune cell subtypes in the tumor microenvironment. The signature gene markers defining each immune cell subtype include CD19 and CD79A for B cells, CD3 and CD4 for CD4+ T cells, and CD8A and CD8B for CD8+ T cells. Macrophages can be identified by the expression of NOS2, IRF5, PTGS2, CD164, VSIG4, and MS4A4A. Neutrophils are characterized by ITGAM and CCR7, and dendritic cells express HLA-DPB1, HLA-DRA, HLA-DPA1, CD1C, NRP1, and ITGAX [32].

### 3.5. Clinical Trials

According to ClinicalTrais.gov, a publicly available online database of clinical trials for a wide range of medical conditions and diseases, several pharmaceutical companies are investigating the anti-cancer effects of drugs targeting CCR8 [33]. Drugs such as BAY3375968, SRF114, S-531011, and GS-1811 are antibody-based drugs blocking CCR8, which is located on the surface of regulatory T cells (Table 7). The rationale for targeting CCR8 in cancer therapy is to induce and enhance anti-tumor immune responses by depleting or suppressing Tregs [2,8]. CCR8 inhibition reduces immunosuppression by suppressing Tregs, and anti-PD-1 drugs enhance the immune response against cancer by promoting the activity of cytotoxic T cells [34]. This combination therapy can inhibit the tumor’s immune evasion mechanisms, resulting in synergistic effects and increasing the possibility of overcoming resistance [8].

### 3.6. Prioritization of Target Indications

Our omics-integrated comprehensive analysis investigating differentially expressed genes, survival prognosis, and relationships with immune infiltrating cells by cancer type, presented in Table 2, provides evidence for the association between CCR8 and specific indications. Based on the overall evaluation criteria mentioned above, it was concluded that anti-cancer targets showing strong and moderate correlations for each item would be reasonable for BRCA, COAD, HNSC, READ, STAD, and THCA indications (Table 8).

## 4. Discussion

Our study demonstrates how to evaluate the genetic expression and immune environment of CCR8 based on a bioinformatic repository and identify relevant target indications. This has expanded our understanding of CCR8’s potential as an anti-cancer target by presenting consistent substantiation from multiple analytical perspectives. The analysis results were consistent with common knowledge that suppressing CCR8 signaling reduces immunosuppressive Treg infiltration and plays an important role in shifting the balance of immune cells into an anti-tumor response [35,36,37,38]. In addition, according to our findings, inhibiting CCR8 specifically increases the infiltration of cytotoxic T cells into the tumor microenvironment, which may generate a more potent anti-tumor immune response. We applied this principle to improve the activation and function of cytotoxic T cells when combined with immune checkpoint inhibitors such as PD-1 or PD-L1, which suppress excessive responses and block signals of the immune system [39,40,41]. This synergistic action helps overcome resistance mechanisms that prevent immune cells from recognizing and attacking tumor cells [38,39]. Research into targeted therapies for chemokine receptors such as CCR8 has also presented challenges in the field of anti-cancer drug research [8].

Mogamulizumab, a monoclonal antibody that targets the CCR4 chemokine receptor that is overexpressed on malignant T cells, has been approved for the treatment of cutaneous T-cell lymphomas (CTCL), specifically mycosis fungoides and Sezary syndrome [42]. The approval of mogamulizumab for CTCL seemed to address an unmet need in a difficult-to-treat type of lymphoma, but it showed limitations in actual clinical practice. A limitation of mogamulizumab is the development of resistance mechanisms, including genetic mutations or changes in the expression level of the target CCR4 receptor [43]. These changes can lead to reduced drug efficacy and may contribute to side effects such as graft-versus-host disease (GvHD) due to impaired immune regulation [44]. In this regard, the challenges we must overcome in the development of CCR8-targeted anti-cancer drugs include toxicity, selectivity issues, short half-life, potential resistance and escape mechanisms, limited penetration into the tumor microenvironment, difficulty in patient selection, and the need for a streamlined dosing approach [45,46]. Overcoming these hurdles is critical to realizing the therapeutic potential of drugs and achieving effective anti-cancer outcomes. It is known that a major factor associated with early resistance to ICI inhibitors is the lack of tumor T cell infiltration, which characterizes “cold tumors” [47]. Cold tumors show low T cell infiltration, lack immune activation signals, and are particularly enriched in regulatory T cells (Tregs) and myeloid-derived suppressor cells (MDSCs), creating an immunosuppressive environment, low mutational burden, and resistance to immunotherapy [48]. Considering these characteristics of these tumors, targeting CCR8 selectively and interfering with the CCR8–CCL1 pathway could be another strategy to induce depletion of Tregs [49].

Our comprehensive analysis approach presents both advantages and limitations. One of its strengths is that it systematically collects omics and survival data from accredited public databases, facilitating early prioritization of target indications during drug development. This has the potential to streamline the early-stage process of drug development, minimizing the need for extensive in vitro and in vivo experiments. Limitations of bioinformatics-based targeted studies arise from factors such as incomplete or inconsistent data, algorithm bias, and complexity of biological systems. Also, interactions between molecules can be overlooked, and the results may not always be reproducible or relevant in real-world situations. For this reason, the identified targets must be validated through subsequent in vitro and in vivo experiments to ensure their reliability and clinical significance.

## 5. Conclusions

In conclusion, our comprehensive multi-omics analysis demonstrates the potential of CCR8 as a novel and promising anti-cancer drug target across step-by-step analyzes. Through an approach that includes systematic data collection, gene function analysis, profiling of differential gene expression, immune cell infiltration, prognosis analysis and strategic prioritization of target indications, we propose a potential cancer therapeutic target for CCR8. The identification of target indications such as BRCA, COAD, HNSC, READ, STAD, and THCA further strengthens the hypothesis that CCR8-targeted therapeutic strategies may be a new option for cancer treatment.

## Figures and Tables

**Figure 1 biomedicines-11-02910-f001:**
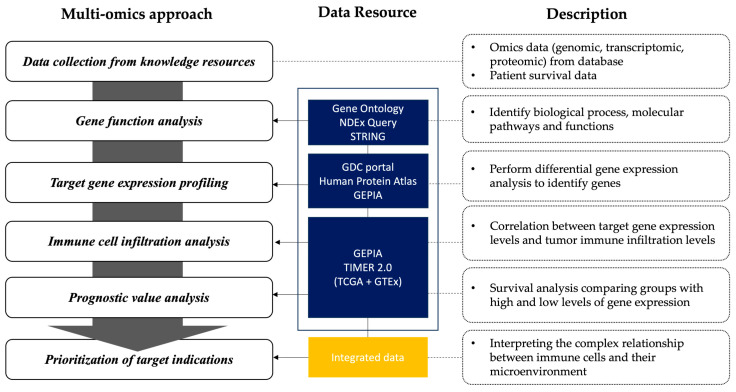
Framework for omics-integrated analysis. This schematic illustrates the comprehensive data analysis flow conducted in the study.

**Figure 2 biomedicines-11-02910-f002:**
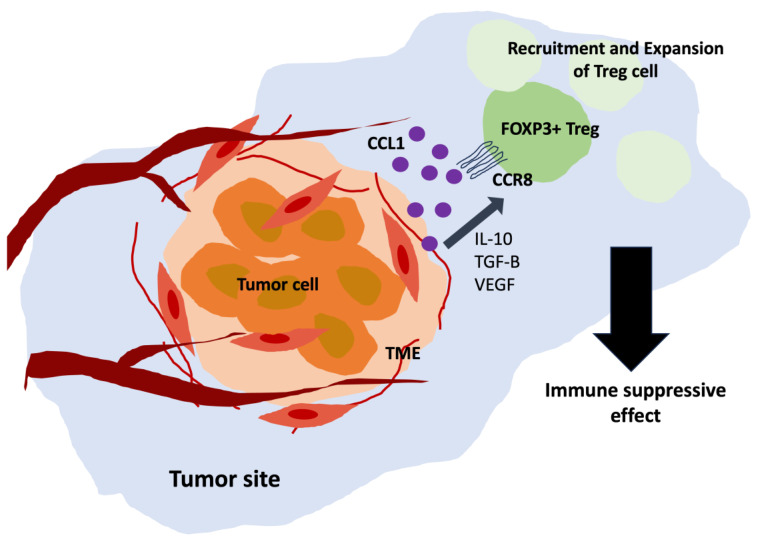
Role of CCR8 in the recruitment and expansion of tumor-infiltrating regulatory T cell (Treg) into the tumor microenvironment (TME). The CCL1–CCR8 axis may contribute to immunosuppression by recruiting Treg to the TME, potentially influencing tumor immune evasion.

**Figure 3 biomedicines-11-02910-f003:**
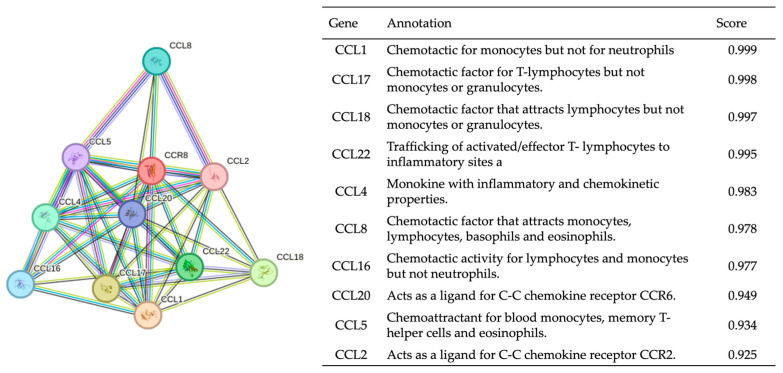
Protein–protein interaction with CCR8. This interconnected network represents a potential correlation between CCR8 and the other related proteins.

**Figure 4 biomedicines-11-02910-f004:**
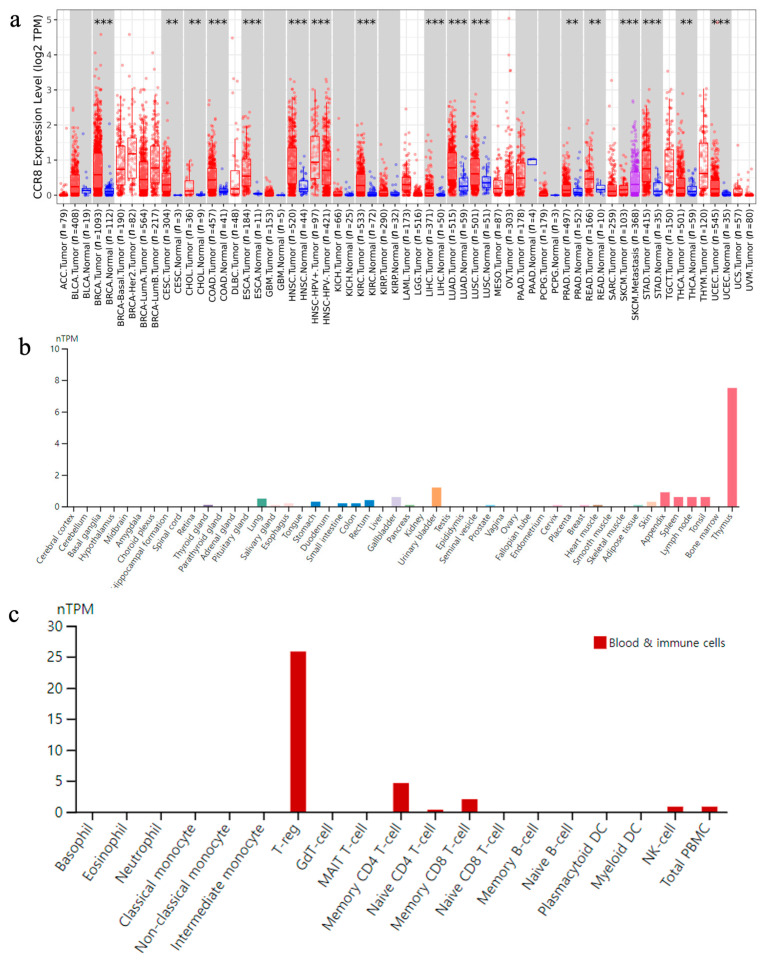
Results of CCR8 gene expression profiling. (**a**) The differential expression of CCR8 between tumor tissues and adjacent normal tissues, categorized by cancer type. The distribution of gene expression levels is depicted using a box plot, with the red box plot representing tumor tissue and the blue one representing normal tissue. The statistical significance computed by the Wilcoxon test is annotated by the number of stars (**: *p*-value <0.01; ***: *p*-value <0.001). (**b**) Overall CCR8 protein expression across 44 organs in normal tissues. (**c**) Results of evaluating CCR8 expression in immune cells infiltrating the tumor microenvironment.

**Figure 5 biomedicines-11-02910-f005:**
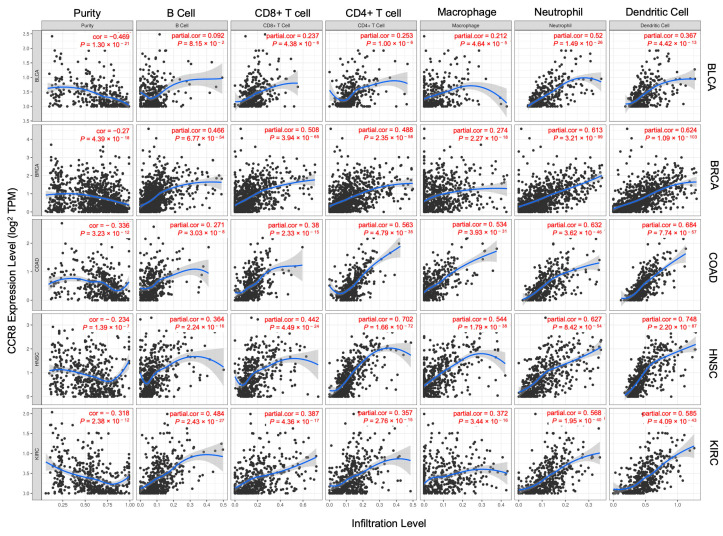
CCR8 expression correlated with immune infiltrating cells in various cancer types. CCR8 shows positive correlations with a variety of immune cell types, including B cells, CD4 T cells, CD8 T cells, neutrophils, macrophages, and dendritic cells. In the graph, dots represent individual data points, lines indicate the overall trend or correlation between immune cell infiltration and target gene expression, while shading denotes the variability or uncertainty around the trend line.

**Figure 6 biomedicines-11-02910-f006:**
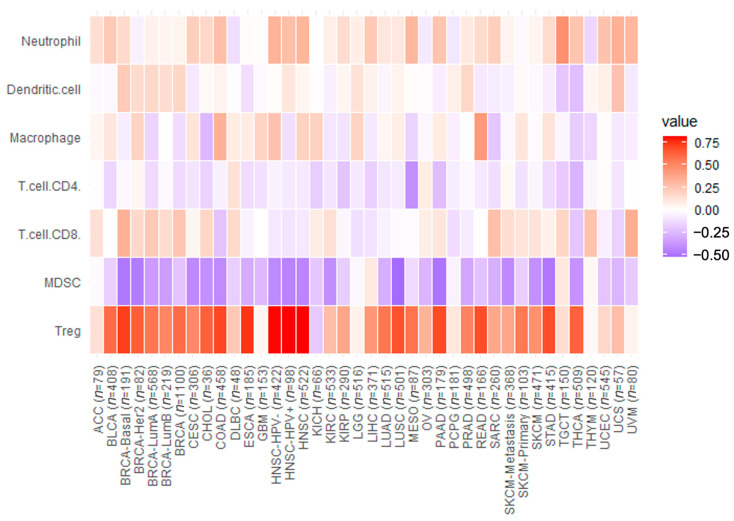
Heatmap of the correlation of CCR8 expression with immune-related cells (Tregs, Myeloid-derived suppressor cells (MDSC), CD8+ T cell, CD4+ T cell, Macrophage, Dendritic cell, Neutrophil). In this analysis, the Treg showed highest correlation with CCR8 expression. Although, neutrophils and dendritic cells showed significant correlation with CCR8, these levels were lower than those for Treg.

**Figure 7 biomedicines-11-02910-f007:**
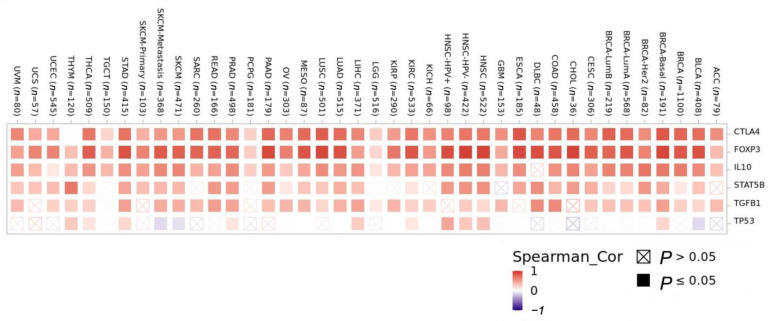
Correlation between Treg gene markers (CTLA4, *FOXP3*, IL-10, STAT5B, TGFB1) and control gene (*TP53*) according to CCR8 expression according to cancer type.

**Figure 8 biomedicines-11-02910-f008:**
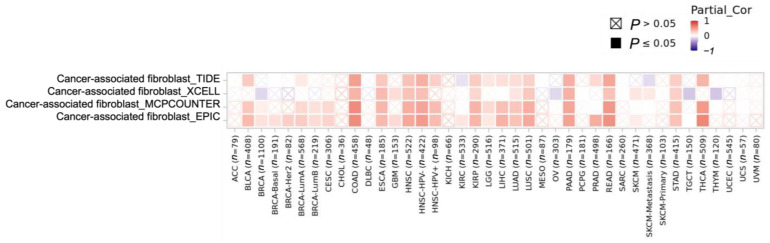
Correlation between CAF levels and CCR8 expression by cancer type. TIDE, XCELL, MCPCOUNTER, and EPIC are tools used to investigate the interaction between cancer-associated fibroblasts (CAFs), specialized cells found in cancer tissue, and specific genes. These tools help researchers understand how CAFs, special cells found in cancer tissue, interact with specific genes. TIDE: TIDE predicts a patient’s immune response to cancer immunotherapy. xCell: xCell assesses the abundance of various cell types in tumor tissue. MCPcounter: MCPcounter provides insight into the immune nature of cancer by quantifying immune and other cell types in the tumor microenvironment. EPIC: EPIC assesses immune pathways and genetic changes to understand immune signatures within cancer tissue.

**Figure 9 biomedicines-11-02910-f009:**
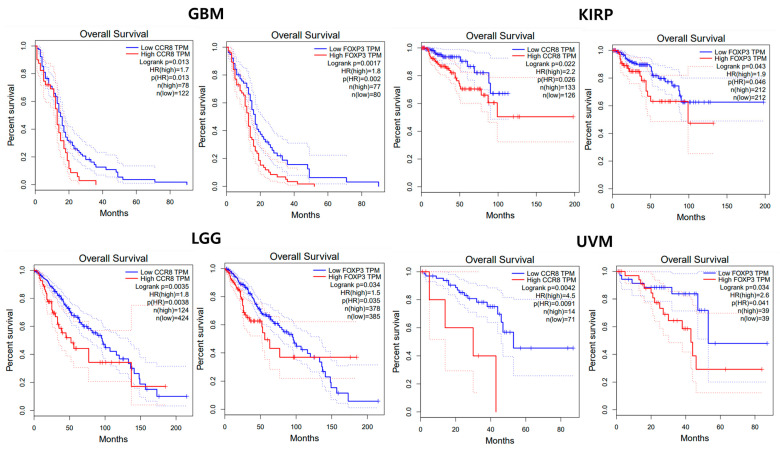
Patient survival analysis based on CCR8 and *FOXP3* expression in different cancers. Kaplan–Meier survival analysis plot showing the impact of CCR8 and *FOXP3* expression on patient survival in four cancer types: GBM, KIRP, LGG, and UVM. Each cancer type is represented by a separate pair of graphs. The dotted line in a Kaplan-Meier survival analysis plot represents censored data points, indicating individuals who have not experienced the event of interest (e.g., death) by the end of the study or at the time of censoring.

**Table 1 biomedicines-11-02910-t001:** TCGA cancer type abbreviations.

Abbreviation	Cancer Type
LAML	Acute Myeloid Leukemia
ACC	Adrenocortical carcinoma
BLCA	Bladder Urothelial Carcinoma
LGG	Brain Lower Grade Glioma
BRCA	Breast invasive carcinoma
CESC	Cervical squamous cell carcinoma and endocervical adenocarcinoma
CHOL	Cholangiocarcinoma
LCML	Chronic Myelogenous Leukemia
COAD	Colon adenocarcinoma
ESCA	Esophageal carcinoma
GBM	Glioblastoma multiforme
HNSC	Head and Neck squamous cell carcinoma
KICH	Kidney Chromophobe
KIRC	Kidney renal clear cell carcinoma
KIRP	Kidney renal papillary cell carcinoma
LIHC	Liver hepatocellular carcinoma
LUAD	Lung adenocarcinoma
LUSC	Lung squamous cell carcinoma
DLBC	Lymphoid Neoplasm Diffuse Large B-cell Lymphoma
MESO	Mesothelioma
MISC	Miscellaneous
OV	Ovarian serous cystadenocarcinoma
PAAD	Pancreatic adenocarcinoma
PCPG	Pheochromocytoma and Paraganglioma
READ	Rectum adenocarcinoma
PRAD	Prostate adenocarcinoma
SARC	Sarcoma
SKCM	Skin Cutaneous Melanoma
STAD	Stomach adenocarcinoma
TGCT	Testicular Germ Cell Tumors
THYM	Thymoma
THCA	Thyroid carcinoma
UCS	Uterine Carcinosarcoma
UCEC	Uterine Corpus Endometrial Carcinoma
UVM	Uveal Melanoma

**Table 2 biomedicines-11-02910-t002:** Criteria for overall evaluation.

Gene Expression Analysis	Immune Cell Infiltration Analysis			Prognostic Value Analysis		Overall Evaluation
	TIICs	Treg	CAF	Uni	Multi	
High expression	Strong correlation in two or more items			Correlation of one or more items		Strong
	Strong correlation in two or more items			Not applicable		Moderate
	Strong correlation in only one items			Not applicable		Low

TIICs, Tumor-Infiltrating Immune Cells; Treg, regulatory T cell; CAF, Cancer-Associated Fibrosis; Uni, Univariate analysis; Multi, Multivariate analysis.

**Table 3 biomedicines-11-02910-t003:** Gene ontology summary of CCR8.

Category Domain	Gene Ontology Term
Biological Process	Immune response
	Cell adhesion
	G protein-coupled receptor signaling pathway
	Chemokine-mediated signaling pathway
	Positive regulation of cytosolic calcium ion concentration
	Chemotaxis
Cellular Component	Plasma membrane
Molecular Function	Coreceptor activity
	C-C chemokine receptor activity
	Chemokine receptor activity

**Table 4 biomedicines-11-02910-t004:** Correlations of CCR8 expression with Treg.

Description	Tregs		Description	Tregs	
	Rho	*p*-Value		Rho	*p*-Value
ACC (*n* = 79)	0.14	0.30	LIHC (*n* = 371)	0.44 ***	<0.001
BLCA (*n* = 408)	0.60 ***	<0.001	LUAD (*n* = 515)	0.56 ***	<0.001
BRCA (*n* = 1100)	0.60 ***	<0.001	LUSC (*n* = 501)	0.68 ***	<0.001
BRCA-Basal (*n* = 191)	0.72 ***	<0.001	MESO (*n* = 87)	0.57 ***	<0.001
BRCA-Her2 (*n* = 82)	0.64 ***	<0.001	OV (*n* = 303)	0.33 **	<0.001
BRCA-LumA (*n* = 568)	0.56 ***	<0.001	PAAD (*n* = 179)	0.69 ***	<0.001
BRCA-LumB (*n* = 219)	0.50 ***	<0.001	PCPG (*n* = 181)	0.11	0.22
CESC (*n* = 306)	0.49 ***	<0.001	PRAD (*n* = 498)	0.51 ***	<0.001
CHOL (*n* = 36)	0.63 ***	<0.001	READ (*n* = 166)	0.68 ***	<0.001
COAD (*n* = 458)	0.70 ***	<0.001	SARC (*n* = 260)	0.37 **	<0.001
DLBC (*n* = 48)	0.23 *	0.19	SKCM (*n* = 471)	0.46 ***	<0.001
ESCA (*n* = 185)	0.74 ***	<0.001	SKCM-Metastasis (*n* = 368)	0.38 **	<0.001
GBM (*n* = 153)	0.04	0.70	SKCM-Primary (*n* = 103)	0.55 ***	<0.001
HNSC (*n* = 522)	0.81 ***	<0.001	STAD (*n* = 415)	0.68 ***	<0.001
HNSC-HPV− (*n* = 422)	0.81 ***	<0.001	TGCT (*n* = 150)	0.14	0.12
HNSC-HPV+ (*n* = 98)	0.81 ***	<0.001	THCA (*n* = 509)	0.62 ***	<0.001
KICH (*n* = 66)	−0.19	0.17	THYM (*n* = 120)	0.03	0.77
KIRC (*n* = 533)	0.28 *	<0.001	UCEC (*n* = 545)	0.17	0.15
KIRP (*n* = 290)	0.38 **	<0.001	UCS (*n* = 57)	0.28	0.07
LGG (*n* = 516)	0.06	0.25	UVM (*n* = 80)	0.05	0.70

*: weak correlation; **: moderate correlation; ***: strong correlation.

**Table 5 biomedicines-11-02910-t005:** Survival prognosis by CCR8 expression level across cancer types. HR: Hazard ratio.

Cancer Type	HR (High vs. Low)	*p*-Value	No. of Patients (High/Low)
ACC	0.90	0.850	19/64
BLCA	0.85	0.430	325/341
BRCA	0.95	0.740	532/533
CESC	0.79	0.320	146/142
CHOL	0.58	0.330	16/16
COAD	0.80	0.370	134/135
DLBC	1.50	0.590	22/21
ESCA	1.20	0.430	91/91
GBM	1.70	0.013 *	78/122
HNSC	0.68	0.019	389/389
KICH	0.89	0.860	29/31
KIRC	1.20	0.230	251/256
KIRP	2.20	0.026 *	133/126
LAML	1.00	0.970	52/50
LGG	1.80	0.004 **	124/424
LIHC	0.83	0.560	244/327
LAUD	0.80	0.150	237/239
LUSC	1.20	0.210	240/236
MESO	1.10	0.810	39/41
OV	0.81	0.095	211/210
PADD	0.84	0.400	87/88
PCPG	2.90	0.240	71/124
PRAD	1.00	1.000	240/233
READ	0.70	0.460	46/46
SARC	0.95	0.810	122/127
SKCM	0.55	1.2 × 10^−5^	225/228
STAD	1.00	0.820	192/192
TGCT	5.63 × 10^8^	1.000	68/67
THCA	1.10	0.840	254/250
THYM	1.30	0.750	58/53
UCEC	1.30	0.480	83/75
UCS	0.64	0.200	25/24
UVM	4.50	0.0091 **	14/71

* *p* < 0.1, ** *p* < 0.05.

**Table 6 biomedicines-11-02910-t006:** Multivariate Cox proportional hazards analysis results.

Covariate	HR (95% CIs)	*p*-Value	Covariate	HR (95% CIs)	*p*-Value
BRCA			COAD		
CCR8 expression	1.29 (1.01–1.64)	0.037 *	CCR8 expression	0.39 (0.17–0.91)	0.029 *
B-cell	0.48 (0.00–34.34)	0.739	B-cell	1.27 (0.01–143.46)	0.920
CD8+ T-cell	0.36 (0.03–3.87)	0.399	CD8+ T-cell	0.02 (0.00–0.98)	0.049 *
CD4+ T-cell	1.19 (0.02–49.38)	0.924	CD4+ T-cell	0.53 (0.00–79.10)	0.804
Macrophage	5.91 (0.41–83.68)	0.188	Macrophage	9.61 (0.08–1149.82)	0.354
Neutrophil	7.32 (0.03–1487.46)	0.463	Neutrophil	0.02 (0.00–46.09)	0.343
Dendritic cell	0.37 (0.04–2.91)	0.347	Dendritic cell	59.39 (2.29–1537.39)	0.014 *
HNSC			Kidney Chromophobe KICH		
CCR8 expression	0.62 (0.43–0.88)	0.009 **	CCR8 expression	1.96 (6.10–6.34)	0.006 **
B-cell	0.10 (0.00–1.77)	0.119	B-cell	9.93 (3.68–2.67)	0.000 ***
CD8+ T-cell	0.24 (0.03–1.77)	0.164	CD8+ T-cell	0.00 (0.00–0.00)	0.000 ***
CD4+ T-cell	0.23 (0.00–6.67)	0.394	CD4+ T-cell	0.00 (0.00–0.00)	0.000 ***
Macrophage	10.98 (0.53–223.99)	0.119	Macrophage	4.68 (3.51–6.23)	0.000 ***
Neutrophil	2.75 (0.11–63.68)	0.528	Neutrophil	4.14 (2.59–6.62)	0.000 ***
Dendritic cell	3.64 (0.74–17.93)	0.112	Dendritic cell	0.00 (0.00–0.00)	0.000 ***
LIHC			MESO		
CCR8 expression	0.51 (0.15–1.79)	0.299	CCR8 expression	0.72 (0.22–2.35)	0.593
B-cell	0.00 (0.00–8.96)	0.175	B-cell	0.15 (0.00–249.47)	0.617
CD8+ T-cell	0.00 (0.00–0.26)	0.012 *	CD8+ T-cell	2.81 (0.01–472.12)	0.692
CD4+ T-cell	0.03 (0.00–30.76)	0.329	CD4+ T-cell	5.97 (0.01–3583.79)	0.584
Macrophage	265.66 (1.17–60,085.23)	0.044 *	Macrophage	5681.43 (3.82–8,434,211.64)	0.020 *
Neutrophil	5.32 (0.00–690,216.97)	0.781	Neutrophil	0.00 (0.00–0.00)	0.000 ***
Dendritic cell	95.65 (2.40–3801.49)	0.015 *	Dendritic cell	389.04 (5.41–27,948.15)	0.006 **
PAAD			OV		
CCR8 expression	0.41 (0.19–8.65)	0.019 *	CCR8 expression	0.94 (0.80–1.12)	0.540
B-cell	7.72 (0.03–1.93)	0.46844	B-cell	0.16 (0.00–64.42)	0.549
CD8+ T-cell	44.50 (0.07–2.84)	0.250	CD8+ T-cell	0.04 (0.00–0.00)	0.000 ***
CD4+ T-cell	0.00 (0.00–2.14)	0.079	CD4+ T-cell	0.00 (0.00–0.00)	0.000 ***
Macrophage	0.01 (0.00–3.26)	0.118	Macrophage	10,467.65 (48.64–2,252,393.25)	0.001 **
Neutrophil	0.01 (0.00–3.26)	0.015 *	Neutrophil	8707.68 (1.58–47,760,091.36)	0.039 *
Dendritic cell	2.76 (0.05–1.32)	0.601	Dendritic cell	0.44 (0.00–44.78)	0.731

*, *p* < 0.05; **, *p* < 0.01; ***, *p* < 0.001. BRCA, Breast invasive carcinoma; COAD, Colon adenocarcinoma; HNSC; Head and Neck squamous cell carcinoma; KICH, Kidney Chromophobe; LIHC, Liver hepatocellular carcinoma; MESO, Mesothelioma; PAAD, Pancreatic adenocarcinoma; OV, Ovarian serous cystadenocarcinoma.

**Table 7 biomedicines-11-02910-t007:** List of CCR8 targeted strategies used in clinical trials.

NCT No.	Drug Name	Combination	Conditions	Types	Status
NCT05537740	BAY3375968	Monotherapy vs. Combination with anti-PD-1	Advanced Solid Tumors	Antibody	Recruiting
NCT05635643	SRF114	Monotherapy	Advanced Solid Tumors and Head and Neck Squamous Cell Carcinoma	Antibody	Recruiting
NCT05101070	S-531011	Monotherapy vs. Combination with anti-PD-1	Advanced Solid Tumors	Antibody	Recruiting
NCT05007782	GS-1811	Monotherapy vs. Combination with anti-PD-1	Advanced Solid Tumors	Antibody	Recruiting

**Table 8 biomedicines-11-02910-t008:** Overall summary evaluation across cancer types. High CCR8 expression coupled with high immune cell infiltration suggests a potential target for clinical trials in specific cancer types.

	Gene Expression Analysis	Immune Cell Infiltration Analysis			Prognostic Value Analysis		Overall Evaluation
		TIICs	Treg	CAF	Uni	Multi	
ACC							
BLCA	**	***	***				
BRCA	***	***	***			**	***
CESC	**		***				
CHOL	**		***				
COAD	***	***	***	**		**	***
DLBC							
ESCA	***		***				*
GBM					**		
HNSC	***	***	***	**		**	***
KICH						**	
KIRC	***				**		
KIRP					**		
LAML							
LGG					**		
LIHC	***		***			**	***
LUAD	***		***				*
LUSC	***		***				*
MESO			***			**	
OV						**	
PAAD	**		***			**	
PCPG							
PRAD			***				
READ	**		***	**			**
SARC							
SKCM	***		***				
STAD	***		***	**			**
TGCT							
THCA	**		***	**			**
THYM							
UCEC	***						
UCS							
UVM					**		

TIICs, Tumor-infiltrating immune cells; Treg, regulatory T cell; CAF, Cancer-associated fibrosis; Uni, univariate analysis; Multi, multivariate analysis. *: low correlation; **: moderate association; ***: strong association.

## Data Availability

The datasets analyzed this study are available in the Gene Ontology database (http://geneontology.org), NDEx Query database (https://www.ndexbio.org/iquery/), STRING database (https://string-db.org), TIMER database (http://timer.cistrome.org), GEPIA database (http://gepia2.cancer-pku.cn/), Human Protein Atlas database (https://www.proteinatlas.org), TCGA database (https://portal.gdc.cancer.gov/).

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
