# Peer review of "CCR8 as a Therapeutic Novel Target: Omics-Integrated Comprehensive Analysis for Systematically Prioritizing Indications"

_biomedicines, 2023, doi:10.3390/biomedicines11112910_

Round 1

Reviewer 1 Report

Authors have performed in silico analysis of CCR8 in various cancer types using publicly available datasets.  They showed that CCR8 expression is correlated with T-regs in the TME, however some control experiments are missing. 

Main comments;

Figure legends do not have any descriptive information. Please include sufficient information legend for each figure and table.

Table 4: It is unclear to me how relevant the comparison of CCR8 with other chemokine ligands is given that there is a very low coexpression profile. Can authors show what other immune molecules coexpress with CCR8?

Line 229: It is not clear to me how the authors created the CCR8 DEG. Were the samples grouped based on CCR8 expression (high vs low)?  If I understand correctly (and if the text describes figure 4?), authors performed DEG between normal and tumor tissues and presented CCR8 expression. If this is the case, authors should revise the text because it is very confusing.

Figure 4 legend: "DEG of CCR8 distribution" does not make sense. Do authors mean Expression of CCR8 across cancer types and cell-type specific expression in immune cells ? Please explain each panel in detail in the legend.

Figure 5 and 6: Can authors explain why authors suddenly focus on T-regs although neutrophils and dendritic cells are showing the highest correlation in figure 5? If CCR8 expression leads to cold TME, how these innate immune cells are enriched in the TME? Authors should clarify this point comprehensively.

Figure 9: As many other immune cells are expressing CCR8 (based on authors' data-figure 5), survival plots by using only CCR8 expression are not very informative. I strongly suggest authors add a second gene (ex. T-reg markers or Foxp3) to justify their claims. Indeed, as a control, other cell type-specific genes should be included to show that T-reg derived CCr8 is the cause of immune suppression.

Indicated in the comments.

Author Response

Dear Reviewer 1,

I would like to thank the reviewer for their thoughtful comments and constructive feedback on our manuscript.

We performed in-silico analysis, which plays an important role in hypothesis generation and target identification in drug discovery. As you mentioned, we agree that experimental validation is essential to confirm the role of CCR8 as a target for anticancer drugs. Our in-silico analysis serves as a preliminary step to identify potential leads for further investigation. We are currently in the process of designing and conducting controlled experiments to validate our findings, and these experiments will be included in our subsequent work.

We ultimately aim to complete the controlled experiments within the next 12 months, and we plan to submit an updated version of the manuscript with the experimental data to address this important aspect of our study.

Once again, we appreciate the reviewer's insights, and we are committed to addressing their concerns to enhance the quality of our study.

Thank you.

Reviewer 2 Report

This research focused on the role of CCR8 in neoplasia. It is a bioinformatics analysis using TCGA omnics data. In general, CCR8 is a marker of Tregs that are an important component of the tumoral immune microenvironment, responsible for the inhibition of the host immune response. In carcinomas, high Tregs have been postulated and in some circumstances confirmed as marker associated to poor prognosis of the tumors. Nevertheless, in hematological neoplasia it is not so clear. This is a pure bioinformatics exercise using gene expression datasets that are publicly available; this is a weak point of the research, as the finding have not been validated using own data. Nevertheless, CCR8 is potentially important marker for drug development. So this findings may be of interest.

Comments/questions:

(1) In Figure 2, it is shown that CCR8 is a receptor for CCL1. Nevertheless, in humans CCR8 is also a receptor for CCL18. Could you please investigate into CCL18 in relationship to Tregs and tumor immunology?

(2) Does CCR8 signalling contribute to Treg cell suppressive function, or potentiate Treg cell proliferation?

(3) Could you please investigate into the expression of CCR8 in Th2 cells, monocytes, and NK cells? Should the expression of CCR8 cells by other immune microenvironment cells should be also taken into account?

(4) Does CCR8-negative Treg exist?

(5) Table 3 shows the GO terms. Nevertheless, the immune suppression function doesn't appear. Usually, GO terms are very broad and only indicative of "general" functions of the genes.

(6) In the introduction, the authors may explain about the development of drugs and targeting drugs. Please refer to https://www.cusabio.com/c-21050.html, and confirm more updated information from reliable sources.

(7) Can tumors also express CCR8?

(8) I you look at the information present in The Human Protein Atlas, the role of CCR8 doesn't look very promising. Not even immunohistochemistry is positive in several tumors, only in thymus there is positivity. Please confirm.

https://www.proteinatlas.org/ENSG00000179934-CCR8

(9) In Figure 3. Is CCR8 inversely associated with Reumatoid arthritis and IL17 pathway?

(10) In the results of section 32. Does the increase of expression of CCR8 reflects and increase of FOXP3-positive Tregs? Are both markers equivalent?

(11) In Table 5, instead of P = 0.00, better use P < 0.001?

(12) In Figure7 and associated text/section/paragraph, did you correlate CCR8 with IL10, CTLA4, and other markers characteritics of Tregs?

(13) How did you identify CAF levels?

(14) In the prognostic analysis using kaplan-meier and log rank test, how did you define the cut-off of gene expression lever of CCR8 to stratify the patients into low and high groups? How many cases per group?

(15) In Table 6. UVM cancer type was significant. Does this subtype of cancer belong to immune priviledged sites?

(16) In the section 3.4.2. How did you defined the different components of the immune microenvironment? What markers defined each covariate\cell subtype?

(17) Line 311. Why did you mentioned "prognosis" regarding the data of table 7. Was any prognostic analysis made?

(18) Table 8 shows a list of drugs and trials. CCR8 is combined with anti PD-1.  What is the rationale of combining with PD-1? Do PD-1 has prognostic relevance in the same analyzed for CCR8 neoplasia?

(19)  Have you analyzed own cases for CCR8? Have you validated CCR8 in own cases, or own tissue samples by immunohistochemistry?

Author Response

Dear Reviewer 2,

We sincerely appreciate your positive feedback regarding the potential importance of CCR8 as a marker for drug development.

We strongly agree with the need for verification using our own experimental data, as you mentioned. The focus of our study was to utilize in silico analysis techniques on publicly available multi-omics data to identify potential candidates for further experimental validation. This study aimed to provide a basis for future studies to further explore the experimental validation of CCR8 as a target in solid tumors. We are currently actively working to obtain our own experimental data to validate the bioinformatic predictions in this study. We would like to emphasize that our intention is to generate insights that can guide future experimental work.

We will endeavor to incorporate your suggestions and concerns into our future research activities.

Thank you.

Round 2

Reviewer 1 Report

Thanks for addressing my comments and clarifying each point.